# Simultaneous Determination and Risk Assessment of Pyrrolizidine Alkaloids in *Artemisia capillaris* Thunb. by UPLC-MS/MS Together with Chemometrics

**DOI:** 10.3390/molecules24061077

**Published:** 2019-03-19

**Authors:** Li-Hua Chen, Jun-Chi Wang, Qi-Lei Guo, Yue Qiao, Hui-Juan Wang, Yong-Hong Liao, Di-An Sun, Jian-Yong Si

**Affiliations:** 1The Key Laboratory of Bioactive Substances and Resources Utilization of Chinese Herbal Medicine, Ministry of Education, Institute of Medicinal Plant Development, Chinese Academy of Medical Sciences & Peking Union Medical College, Beijing 100193, China; lihuachen0706@163.com (L.-H.C.); jcwang@implad.ac.cn (J.-C.W.); MOON100107qy@163.com (Y.Q.); whj200428@163.com (H.-J.W.); yhliao@implad.ac.cn (Y.-H.L.); dasun@implad.ac.cn (D.-A.S.); 2Agilent Technologies Co. Ltd. (China), No.3, Wang Jing Bei Road, Chao Yang District, Beijing 100102, China; guo-ql@163.com

**Keywords:** pyrrolizidine alkaloids, *Artemisia capillaris* Thunb., UPLC-MS/MS, PCX solid-phase extraction, risk assessment, chemometrics

## Abstract

Pyrrolizidine alkaloids (PAs) are natural toxins found in some genera of the family Asteraceae. However, it has not been reported whether PAs are present in the widely used Asteraceae plant *Artemisia capillaris* Thunb. (*A. capillaris*). The purpose of this study was to establish a sensitive and rapid UPLC-MS/MS method together with chemometrics analysis for simultaneous determination and risk assessment of PAs in *A. capillaris*. The developed UPLC-MS/MS method was validated and was confirmed to display desirable high selectivity, precision and accuracy. Risk assessment was conducted according to the European Medicines Agency (EMA) guideline. Chemometrics analysis was performed with hierarchical clustering analysis and principal component analysis to characterize the differences between PAs of *A. capillaris*. Finally, PAs were found in 29 out of 30 samples and at least two were detected in each sample, besides, more than half of the samples exceeded the EMA baseline. Nevertheless, the chemometrics results suggested that the PAs contents of *A. capillaris* from different sources varied significantly. The method was successfully applied to the detection and risk evaluation of PAs-containing *A. capillaris* for the first time. This study should provide a meaningful reference for the rational and safe use of *A. capillaris*.

## 1. Introduction

The pyrrolizidine alkaloids (PAs) are widespread plant secondary metabolites. PAs are regarded as natural toxins on account of their potential health risks for both livestock and human [1,2,3]. Amounts of in vivo and in vitro experiments have shown that PAs principally caused damage to the liver [4,5], some might harm the lungs via the blood vessels [6] and even damage the brain [7]. They can also form a series of DNA adducts that cause genotoxicity [8]. It is reported that the toxicity of PAs is mainly related to their chemical structures. The toxic PAs are composed of an 1,2-unsaturated necine base and a necic acid with a branched chain, whereby the carboxylic ester forms a monoester or a diester group, as does the macrocyclic structure (Figure 1A) [9,10]. In general, the chemical structures of toxic PAs are classified into retronecine-, otonecine- and heliotridine-types according to their necine bases (Figure 1B) [11]. As we know, PAs are accumulated primarily in the form of pyrrolizidine alkaloid N-oxides (PANOs) in many plants [12].

It is reported that no less than 660 PAs have been identified in over 6000 species, accounting for 3% of all flowering plants [13,14]. PAs are mainly distributed in these three families: Asteraceae, Boraginaceae and Fabaceae [13,15,16]. Of the family Asteraceae, the majority exist in the tribes Senecioneae, Eupatorieae and Ageratum, such as *Senecio scandens*, *Eupatorium cannabinum* and *Ageratum conyzoides* [17]. To date, high levels of PAs have been reported in many herbal drugs [18,19,20], hence, our attention was drawn to the other genera of the Asteraceae family. *Artemisia capillaris* Thunb. (*A. capillaris*) has been widely used in the herbal field as a classical Asteraceae family and edible wild plant medicine [21]. For this use it should be harvested along with its aerial parts during certain seasons. Chemical studies have suggested that the major constituents of *A. capillaris* were flavonoids, coumarins, phenolic acids and essential oil [22,23,24], but the existence of alkaloids such as PAs has not been proved. Considering that the occurrence of alkaloids in similar plants of the genus Artemisia like *Artemisia nilagirica* and *Artemisia asiatica* has been revealed [25], it is possible that PAs and other alkaloids are present in *A capillaris*. However, as far as we know, no detailed quantitative study for alkaloids in *A. capillaris* has been reported and there is no official statement on potential PAs intake in China as yet. Accordingly, in order to verify the presence of natural PA toxins in *A. capillaris*, it is necessary to develop a feasible quantitative analysis method and assess the risk based on the total PA contents.

Based on the structure and molecular characteristics of PAs, relevant detection methods with low limits of quantification are important for PA control. According to the published works, a lot of analysis techniques were developed for the analytical study of PAs based on HPLC, LC-MS, GC-MS, etc., [26,27,28]. Typically, the LC-MS methodology was used to collect a large amount of analytical data on PA contamination in food and feed by the European Food Safety Authority (EFSA) Panel [29,30]. However, some isomeric PAs were not baseline separated by this method. So far the ultra-high performance liquid chromatography tandem mass spectrometry (UPLC-MS/ MS) technology has been successfully applied to improve the separation efficiency and sensitivity, meanwhile decreasing the data acquisition time [31]. At the same time, corresponding pretreatment methods have also been widely studied. For example, solid-phase extraction (SPE) or QuEChERS [32] were selected as the purification methods for the detection of PAs in the reported studies. Different types of SPE cartridges were widely used in sample clean-up, such as Bond Elut SCX or Strata-X-C SCX cartridges [33,34,35], whilst some SPE clean-up of herbal products or honey were carried out with C_18_ SPE cartridges [30,36]. In conclusion, the UPLC-MS/MS technology is considered as a good method for PAs detection due to its high separation efficiency. Given that the high requirements for sample purity by UPLC-MS/MS method for the first PAs detection of *A. capillaris*, it is of great significance to develop corresponding pretreatment methods.

In addition, the risk assessment of PAs has been extensively studied. The WHO concluded PAs at daily intakes exceeding 15 μg/kg body weight (b.w.) would lead to hepatotoxicity [37]. The recommended daily intakes of PAs in Australia [38] and The Netherlands [39] are 1 μg/kg b.w. and 0.1 μg/kg b.w., respectively. The UK Committee on Toxicity of Chemicals in Food, Consumer Products and the Environment Committee (COT) estimated that PA dosages of 0.007 μg/kg b.w. per day were less likely to cause worry about cancer risk by applying a margin of exposure of at least 10,000 [40]. Similarly, The German Federal Institute for Risk Assessment (BfR) identified that a daily intake for PAs of 7 ng/kg should not be exceeded [41]. Given the above risk assessments made by the COT and BfR studies, the European Medicines Agency (EMA) suggested the acceptable daily intake of toxic PAs was 0.35 μg for a 50 kg person [42]. Therefore, considering the comprehensiveness of the EMA proposal, this study adopts EMA’s calculated method as the risk assessment.

In this study, ultra-high performance liquid chromatography tandem mass spectrometry (UPLC-MS/MS) coupled with multiple reaction monitoring (MRM) was applied to simultaneously detect PAs in *A. capillaris*. The sample preparation method was optimized to suit the extraction and purification of 30 batches of *A. capillaris* samples. Furthermore, the daily intake of PAs in *A. capillaris* was calculated based on the dose of the Chinese Pharmacopoeia, and a preliminary risk assessment was carried out on PAs in herbal samples according to the EMA recommendation [42]. Finally, to characterize the detected PAs of *A. capillaris* samples, chemometric analysis was performed with hierarchical clustering analysis (HCA) in the form of clustering heatmaps [43] and principal component analysis (PCA). It is expected that the newly established UPLC-MS/MS approach and the optimized pretreatment method, together with the statistical chemometrics analysis could be extended to the determination and risk assessment of other PA-containing herbs, which would provide an important basis for their rational and safe utilization.

## 2. Results and Discussion

### 2.1. Pretreatment Method Development

There is no doubt that the extraction conditions can directly affect the stability of PAs, PA yields, and the ratio of PAs to PANOs, and besides, recovery rates and content differences of PAs are reflected in different brands or materials of SPE cartridges [29,44]. Hence, appropriate extraction conditions and a purification process were further optimized for sample pre-treatment in this study. 

#### 2.1.1. Extraction Optimization

Based on the alkalinity, polarity and existence form of alkaloids, a variety of solvents have been commonly used for their extraction, such as methanol or alcohol, acid-water and lipophilic organic solvent. In order to further optimize extraction conditions, *A. capillaris* samples were extracted from five different solvents: methanol, 0.05 M sulfuric acid in methanol, 0.05 M sulfuric acid water, ethanol and 0.05 M sulfuric acid in ethanol. Each test with different extraction solvents was repeated three times, and the standard deviation (SD) represents the deviation of three parallel samples. When compared with the average total PAs concentrations obtained with the different solvents, methanol gave the highest sample extraction efficiency with an average extraction amount of 133.4 μg/kg, 0.05 M sulfuric acid in methanol came next and 0.05 M sulfuric acid ranked third, where the efficiency of methanol is nearly twice that of sulfuric acid-water, see Figure 2.

In this study, various extraction aids such as sonication, cold soaking and solvent refluxing were compared. While the optimum highest efficiency for PA extraction was solvent refluxing, certain PAs, such as intermedine and its N-oxide, or lycopsamine and its N-oxide, were greatly affected by the temperature, so they were unstable. Sonication had an extraction efficiency very close to that of cold soaking, so it was selected as the preferred extraction mode due to its quick and simple operation characteristics (the details not given). By investigating the frequency and time of extraction, we finally decided that the optimal ultrasonic conditions were one extraction with a power setting of 100 W and ultrasonic frequency of 40 kHZ at 25 °C for 30 min.

#### 2.1.2. Purification Optimization

The selection of a suitable SPE cartridge has a significant effect on the efficiency of PA capture and subsequent recovery during sample preparation. In this study, five different varieties of SPE cartridges (500 mg/6 mL) were investigated: Cleanert PCX, SCX, C_18_, C_8_/SCX (all from Agela Technologies) and Strata-X-C (Phenomenex). 

When a blank mixed acid standard solution without a matrix was added to each of the five SPE cartridges, the recoveries of these 34 compounds with Cleanert PCX SPE cartridges ranged from 68.42% to 102.08%. However, the recoveries with Cleanert SCX and C_8_/SCX fluctuated widely and those of several PAs were only 60% or less. Although the recoveries of C_18_ and Phenomenex Strata-X-C was relatively stable on the whole, for some individual PAs they were less than 60%, as shown in Figure 3 (detailed recoveries are listed in the Appendix A). To sum up, Cleanert PCX SPE, a mixed-mode strong cation exchange sorbent, was found to be the most appropriate clean-up cartridge. It can provide dual ion exchange and reverse-phase retention modes. Furthermore, different ratios of the elution solvent NH_4_OH/MeOH (3:17→1:3, *v/v*) were tested to determine the highest elution rate of PAs (details are not displayed). The results showed that the highest elution yield of total PAs was obtained by using a 1:3 (*v/v*) solution of NH_4_OH/ MeOH.

#### 2.1.3. Redissolution Optimization

Most alkaloids are lipophilic, and PAs are no exception, whilst PANOs are more hydrophilic and exhibit a higher water solubility than PAs. Therefore, three different ratios of methanol/water were estimated: pure methanol, methanol/water (50/50, *v/v*) and methanol/water (5/95, *v/v*). The results revealed that the recoveries and solubility were relatively high when the residue was dissolved in methanol/water (50/50, *v/v*).

### 2.2. UPLC-MS/MS Method Development

Coupling of UPLC with MS/MS instruments for the determination of PAs has been proved to achieve satisfactory results in this paper (Appendix A). Under optimum conditions (see Section 3.3), 18 PAs and 14 PANOs, including nine sets of isomers were separated completely among these 34 compounds. 

Only two pairs of stereoisomers (intermedine and indicine, intermedine N-oxides and indicine N-oxides) were not separated, but the two unseparated PAs had the same response value as their corresponding isomers. The MRM chromatograms of these isomers are shown in Figure 4. The application of UPLC successfully improved the chromatographic resolution and sensitivity, and dramatically reduced the acquisition time. In the analysis, two organic solvents, methanol and acetonitrile, were compared, it was found that the peak shape and separation effect of methanol was better. By optimization, methanol and water with 0.05% formic acid and 2.5 mM/L ammonium formate were selected as acidic mobile phases. 

Besides, different kinds of columns, including the Agilent Zorbax Eclipse Plus C_18_ column (150 × 3.0 mm, 1.8 μm), Waters Acquity BEH C_18_ column (100 × 2.1 mm, 1.7 μm) and Phenomenex Kinetex EVO C_18_ column (150 × 2.1 mm, 2.6 μm), were evaluated. Results suggested that the Phenomenex C_18_ column would shorten the analysis time, but the mixed reference solutions couldn’t achieve satisfactory separation compared to the Zorbax C_18_ column (shown in Figure 5). Additionally the UPLC binary gradient pump was unable to meet the high pressure demand of the BEH C_18_ column under the chromatographic conditions. Thus, the Zorbax Eclipse Plus C_18_ column was applied in the following experiments. 

Due to the polarity of PAs, they give a better results by utilizing electrospray ionization (ESI) in positive mode, which is more appropriate than APCI for testing polar compounds. In the MS experiments, the protonated molecule [M + H] ^+^ of each PA was visible in the complete MS spectrum. As for the MS^2^ spectra, fragment ions of *m/z* 120, 118, 138, 136, 150 and 168 are characteristic for PA free bases related to necine, while product ions of *m/z* 120 and 138 are typical of retronecine-type and heliotridine-type PAs, and *m/z* 168 and 150 for otonecine type. Precursor ions, product ions, fragmentor and collision energy were automatically optimized in the ESI ( + ) ion with MRM mode. The MS conditions of 34 PAs and the corresponding N-oxides are listed in Appendix A. 

### 2.3. Method Validation

#### 2.3.1. Sensitivity, Linearity, LOD and LOQ

The calibration curve of a wide concentration range from 0.1 ng/mL to 500 ng/mL for the analyses of PAs in herbal plants was verified by fortifying standard substance mixtures. Among these 34 PAs, the limit of detection (LOD) and the limit of quantification (LOQ) values of each individual PA were in the range of 0.01~0.2 μg/kg and 0.1~0.5 μg/kg, respectively. A detailed listing of the limits of detection and limits of quantification is provided in Table 1.

#### 2.3.2. Precision and Recovery

In the instrument performance test, the tolerance of repeatability and stability behaved well with the RSD < 5% by repeated injection of 100 ng/mL reference standards (n = 6). The intra-day and inter-day repeatability expressed by RSD (%) were less than 8% and no significant distinction was found between them. The mean recoveries of spiked samples with mixed standards injection in triplicate were within the range of 68–103%. The precision and recovery information is detailed in Table 1.

#### 2.3.3. Matrix Effects

Since the matrix (unlike analytes) often causes significant interferences during the analysis process and affects the accuracy, these influences and interferences are called matrix effects, and may result in ion enhancement or ion suppression during the LC-ESI-MS/MS analysis [45]. Mixed standards was added to the blank matrix of *A. capillaris* samples purchased from Zhengzhou City, Consequently, all components but retronecine were found to meet the requirements (85%~115%), as exhibited in Table 1.

### 2.4. The Occurrence of PAs in A. capillaris Samples

Based on the UPLC-MS/MS method established above, the contents of *A. capillaris* samples from 30 different regions were determined. In order to summarize the occurrence of PAs types in each sample, the distribution of each major detected PA in 30 tested *A. capillaris* samples are exhibited in Figure 6A. In a total of twenty nine (29) samples among the 30 analyzed (96.7%) by UPLC-MS/MS, at least two PAs were quantified, that is to say, PAs were detected in all of the *A. capillaris* samples except that purchased from Zhengzhou City. In these 29 samples, eight (8) components were detected out of the studied 34 PAs, and they were intermedine (Im), lycopsamine (Ly), seneciphylline (Sp) and their N-oxides (ImNO, LyNO and SpNO), senkirkine (Sk) and echimidine N-oxide (EmNO). Sp, SpNO and EmNO are not displayed in Figure 6A, because they were only found in small amounts in one or two plants. The chemical structures of five detected PAs in most samples are shown in Figure 6B. The other PAs studied were not present or were below the quantification limits. The result was consistent with that reported in a previous reference [46], which revealed that these eight PAs occurred in high frequency in herbal infusions. In contrast, lasiocarpine, the most toxic PA [29], had not been detected in any sample of the test. 

The concentration of each individual PA and the total contents in *A. capillaris*. samples are shown in Appendix A The total contents in these PA-positive samples ranged from 0.39 to 2111.22 μg/kg. With regards to the total concentrations, among these 30 samples, 13 were less than 10 μg/kg, 11 samples were between 10 and 100 μg/kg, the other six were more than 100 μg/kg, so it seems that the differences might be related to the sources, harvesting time and regional distribution. In addition, the occurrence of PAs of *A. capillaris* could be the result of accidental contamination of PA-containing herbs or they existed as endogenous products. In order to ensure the safety of use, further research in terms of the relation between the presence of PAs and *A. capillaris* sources and the cause of PAs exposure is necessary.

To calculate the probability of the same PAs being present in the 30 samples, Table 2 summarizes the frequency and the concentration range of individual PAs present in the *A. capillaris* samples. It can be roughly inferred from the table that the first five PAs might be endogenous substances. LyNO showed the highest occurrence (up to 96.7%) with PA levels ranging from 0.21 to 1750.99 μg/kg, Im and ImNO were also found to have a relatively high incidence (both accounting for 83.3%) with concentrations ranging from 0.11 to 383.28 μg/kg and from 0.20 to 255.46 μg/kg, respectively. These were followed closely by Ly and Sk (both 73.3%) with PA contents that varied from 0.29 to 92.05 μg/kg and from 0.10 to 15.81 μg/kg, respectively. In general, Im, Ly and LyNO were the major contributors to the total PA contents, with the highest contribution rates reaching 79.1%, 82.6% and 82.9%, respectively. 

### 2.5. Risk Assessment for the intakes of PAs in A. capillaris Samples

Risk assessment related to the toxic PAs was calculated by the EMA statement combined with the COT and BfR studies [47]. Based on this approach, the tolerable levels of exposure for PAs should not exceed 0.007 μg/kg b.w. per day, and 0.35 μg/day was defined as the baseline value of risk assessment for adults with a body weight of 50 kg. It should be indicated that the provisional daily intake of PAs from *A. capillaris* in this research is only for reference orientation purposes. According to the provision in the Chinese Pharmacopoeia (2015), the daily dosage of *A. capillaris* (dry weight) is 6~15 g. Accordingly, it was calculated by the herbal consumption that the minimum daily intake and the maximum daily intake of PAs in 29 PA-positive samples (the sample Y20 from Zhengzhou City was not included) ranged from 0.002 to 12.67 μg and from 0.006 to 31.67 μg, respectively. The risk assessment of PAs were calculated using the maximum daily intakes, and the ones exceeding the baseline value are shown in Figure 7. 

After the risk assessment of the above 30 batches of *A. capillaris* samples, it was found that 10 samples exceeded the PAs baseline at the lowest daily dosage and 16 at the highest daily dose, respectively. From the chart, the daily intake of PAs from sample Y22 was more than 95 times that of the baseline in the sample containing the highest content, while sample Y4 came next at 35 times (12.28 μg). On average, there were also four samples (samples Y3, Y24, Y29 and Y30) where the maximum daily intakes were more than five times higher than the baseline. The remaining samples all slightly exceeded the baseline (but by no more than five times).

### 2.6. Chemometrics Analysis of PAs in A. capillaris Samples

#### 2.6.1. Hierarchical Clustering Analysis

HCA, a multivariate analysis technique, is commonly used to classify samples into groups by generating dendrograms [48]. A hierarchical clustering heatmap was used in this study to visualize the relations between different sample sources and PA concentrations (Figure 8). As shown in the HCA heatmap, the *A. capillaris* samples were roughly classified into two main groups (I and II). Group I (in red font) consisted of samples with total PA contents exceeding the baseline, which included 16 samples (Y1–Y5, Y7–Y9, Y22–Y24 and Y26–Y30). Group II (in black font and in the green box) was composed of samples with PA levels not exceeding the baseline. This result was in accordance with that of the daily intake given in Figure 7. 

Among them, Im, Ly, ImNO and LyNO all contributed to the amounts of PAs in sample Y22 from Gansu Province and sample Y29 from Jiangxi Province. However, in samples Y1 to Y3 and sample Y7, which were obtained from North China, as well as sample Y26 (from East China) and sample Y5 (from Hunan Province), the high PAs contents was attributed to ImNO and LyNO. In addition, LyNO was the uppermost contributors to the high PA concentrations in samples from East China (Y4, Y24 and Y30) and Y8 (from Shaanxi Province). Finally, ImNO was the dominant contributor to the sample from Hubei Province (Y23) and samples from East China (Y27 and Y28), while Im contributed mainly to sample Y9.

#### 2.6.2. Principal Component Analysis

The PCA method was applied to describe the characteristics of 16 batches of *A. capillaris* samples with the PAs daily intake above the baseline. Figure 9 illustrated the significant differences of PAs between samples from various regions. Five main detected PAs (Im, ImNO, Ly, LyNO and Sk) in the samples were set as variables during this evaluation. Results displayed that the first principal component (PC1) and the second principal component (PC2) accounted for 57.2% and 21.7%, respectively (Figure 9A). It was seen that the cumulative percentage of variance by PC1 and PC2 was 78.9%, indicating the deviations of PAs of *A. capillaris* samples from different regions. 

Among these samples with excessive PAs levels, six samples (Y3, Y4, Y22, Y24, Y29 and Y30) with daily intakes exceeding the baseline and four (Y4, Y24, Y29 and Y30) of the samples derived from East China, sample Y3 from North China and sample Y22 from the Northwest, respectively, showed significant PA variability. By contrast, no remarkable difference was observed in the other 12 samples surpassing the baseline slightly. From the loading plots of five dominant detected PAs (Figure 9B), ImNO or LyNO, the main contributors to samples Y4, Y24, Y30 and Y22, correlated positively with PC1 and PC2. Similarly, Im correlated positively with PC1, illustating the distribution characteristics of sample Y29. Sk was negatively correlated to PC1, while it displayed a positive correlation with PC2. Compared with other samples, high content of Sk occurred in sample Y3, which resulted in the significant deviation of Y3.

## 3. Materials and Methods

### 3.1. Chemicals and Reagents

Reference standards of retronecine (Ret, **1**), echimidine (Em, **2**), echimidine N-oxide (EmNO, **3**), erucifoline (Er, **4**), erucifoline N-oxide (ErNO, **5**), europine (Eu, **6**), europine N-oxide (EuNO, **7**), heliotrine (He, **8**), heliotrine N-oxide (HeNO, **9**), intermedine (Im, **10**), intermedine N-oxide (ImNO, **11**), jacobine (Jb, **12**), jacobine N-oxide (JbNO, **13**), lasiocarpine (Lc, **14**), lasiocarpine N-oxide (LcNO, **15**), lycopsamine (Ly, **16**), lycopsamine N-oxide (LyNO, **17**), monocrotaline (Mc, **18**), monocrotaline N-oxide (McNO, **19**), retrorsine (Re, **20**), retrorsine N-oxide (ReNO, **21**), senecionine (Sn, **22**), seneconine N-oxide (SnNO, **23**), seneciphylline (Sp, **24**), seneciphylline N-oxide (SpNO, **25**), senecivernine (Sv, **26**), senecivernine N-oxide (SvNO, **27**), indicine (Ic, **28**), indicine N-oxide (IcNO, **29**), 7-acetylintermedine (7-Im, **30**), 7-acetylintermedine N-oxide (7-ImNO, **31**), senkirkine (Sk, **32**), trichodesmine (Td, **33**) and 7-acetyllycopsamine (7-Ly, **34**) were all purchased from Phytolab (Vestenbergsgreuth, Germany), the structures of these compounds are shown in Appendix A.

Methanol (MeOH, UPLC/MS grade) was purchased from Thermo Fisher Scientific (Shanghai, China) and water from Wahaha Company (Hangzhou, China). Formic acid (HPLC grade) was obtained from DIKMA Technologies Inc. (Lake Forest, IL, USA) and ammonium bicarbonate (99% purity, HPLC grade) from MREDA Technologies Inc. (Beijing, China). Both ammonium hydroxide in water (NH_4_OH, 25%) and sulfuric acid (H_2_SO4, 98%) were purchased from Beijing Chemical Works (Beijing, China).

*A. capillaris* samples (n = 30) deriving from 21 different provinces were purchased from several medicine markets and pharmacies. Among these samples, four were from Zhejiang Province, the number of samples coming from Shanxi and Hebei provinces was three and from Shandong and Hubei province there were two, and the number of samples from other provinces was only one. They were identified by Dr. Yu-Lin Lin of Institute of Medicinal Plant Development in Chinese Academy of Medical Sciences (Beijing, China). Voucher specimens were depostited at the Institute of Medicinal Plant Development (Peking Union Medical College, Beijing, China). A summary of the source details is given in Appendix A.

### 3.2. Standard Solutions and Sample Preparation

#### 3.2.1. Standard Solutions

A stock solution of each pure PA standard was prepared at a concentration of 100 μg/mL in acetonitrile and stored at −20 °C. To obtain a concentration of 1 μg/mL standard working solution, respective volumes of each PA stock solution are combined into mixed reference standards in methanol (HPLC grade) and stored at 4 °C after serial dilution. 

#### 3.2.2. Sample Preparation

*A. capillaris* samples were ground using a grinder and the resulting powder was separately sieved through 60 mesh (0.3 mm). Each homogeneous sample was mixed using a shaker, and a 2.0000 ± 0.0005 g portion was accurately weighed and transferred to a flask with 40 mL of MeOH. Extraction was performed in an ultrasonic bath for 30 min at room temperature. All samples were centrifuged at 6000 rpm (4430× *g*) for 10 min. The supernatant was then transferred to a beaker (50 mL) and concentrated up to dryness followed by the PCX-SPE procedure.

#### 3.2.3. PCX-SPE Procedure

The PCX-SPE cartridges were preconditioned with 5 mL of MeOH followed by 5 mL of 0.05 M sulfuric acid. After the sample was loaded, the cartridge was washed with 5 mL of 0.05 M H_2_SO_4_ and then with 10 mL of MeOH. The target PAs compounds were eluted using 10 mL of NH_4_OH/MeOH solution (1:3, *v/v*), which should be freshly prepared per working day. After the eluted samples dried at 50 °C under nitrogen, they were reconstituted in 2 mL of methanol/water (50:50, *v/v*) and then directly filtered into an amber LC vial (2 mL) using a syringe filter (0.22 μm).

### 3.3. Ultra-High Performance Liquid Chromatography-Mass Spectrometry (UPLC-MS/MS) Analysis

All herbal samples were separated and analyzed using an Agilent 6470 triple quadrupole mass spectrometer with Agilent Jet Stream technology [49] in ESI positive ionization mode and an Agilent Infinity II 1290 UPLC system (Agilent Technologies, Santa Clara, CA, USA). The chromatographic separation was performed with a Zorbax Eclipse Plus C_18_ column (3.0 mm × 150 mm, 1.8 μm, p/n 959759-302; Agilent Technologies Inc.), and the column was maintained at 40 °C (± 0.8 °C). The mobile phases consisted of solvent A (water) and solvent B (methanol), both mixture with 0.05% formic acid and 2.5 mM/L ammonium formate. A binary gradient profile was achieved as follows: 0.5 min, 5% B; 0.5–1.0 min, 5–20% B; 1.0–11.0 min, 20–37% B; 11.0–13.0 min linear increased in B from 37% to 95%; 13.0–15.5 min held at 95% B and 16.0 min returned to 5% B, re-equilibration time between each run was 3.0 min. The injection volume was 2 μL and the flow rate was maintained at 0.40 mL/min. The chromatography system configuration and parameters are summarized in Appendix A.

The mass spectrometer was performed in the positive-ion mode of the ESI source using Agilent Jet Stream technology and the following parameters was created: drying gas temperature, 300 °C; drying gas flow, 7.0 L/min; nebulizer pressure, 40 psi; sheath gas heater, 325 °C; sheath gas flow, 11L/min and capillary voltage, 3500 V. Nitrogen was used as the drying and sheath gas.

Multiple reaction monitoring (MRM) mode provides a highly sensitive and selective method for simultaneous quantitative analysis of specific compounds in this experiment. The MRM values for all scan transitions were as follows: time filter width, 0.07 min; dwell time, 5 ms; and Delta EMV, 300V. Appendix A details the mass spectrometer instrument settings and parameters. MS/MS spectra were obtained by the infusion of 100 ng/mL reference solutions of the targeted compounds, so as to determine optimal precursor and product ions, fragmentor voltages, and collision energies.

### 3.4. Method Validation

The analytical method was validated in terms of the linearity, repeatability (precision), recovery (accuracy), limits of detection (LOD), and limits of quantification (LOQ) according to Commission Decision No.2002/657/EC. Linearity was determined by three sets of calibration curves which were divided into two different concentrations: seven-point linear range from 0.1 ng/mL to 10 ng/mL for low level compounds and the range from 10 ng/mL to 500 ng/mL for the high level. The *A. capillaris* sample purchased from Zhengzhou City was chosen for spike experiments. The recovery of PAs was evaluated by analyzing triplicate samples at three different concentration levels (low, medium and high) of mixed standards adding to blank samples. The precision was performed by using the same spiked samples based on six replicated injections and was calculated in the form of RSD values. Intra-day variability analysis was tested in triplicate in a single day, whereas inter-day repeatability was analyzed in three different consecutive days. The LOD and LOQ were estimated at signal-to- noise (S/N) ratios of 3 and 10, respectively.

### 3.5. Method Acquisition and Statistical Analysis

Data acquisition was controlled by Agilent MassHunter Acquisition Software (B.08.00), and MS/MS transitions were performed using Agilent MassHunter Acquisition optimizer software. Data were processed with Quantitative Analysis Software (B.08.00) and Qualitative Analysis Software (B.07.00), respectively. The classification of *A. capillaris* samples were performed by HCA in the form of the clustering heatmap, which was conducted using with HemI (Heatmap Illustrator, version 1.0, China), a novel software package, to exhibit the differences of PAs with various sample sources [50,51]. The squared Euclidean distance was used as the metric in the clustering approaches. PCA, a multivariate statistical method to select the principal components representing most of the original variable information [52], was carried out using OriginPro 2018 SR1 (OriginLab Inc., Northampton, MA, USA).

## 4. Conclusions

This study combined UPLC-MS/MS and chemometrics methods innovatively to achieve the simultaneous detection and risk assessment of *A. capillaris* PAs. The optimized pretreatment and UPLC-MS/MS method has been demonstrated to obtain high extraction rate and sensitive analysis effects with acceptable method validation parameters (linearity, LOD, LOQ, precision, recovery and matrix effect). *A. capillaris* from different regions were analyzed by UPLC-MS/MS and chemometrics methods, it was found that 29 of the 30 samples contained PAs and at least two PAs were detected, the chemical structures of most PAs were Im and Ly with their corresponding N-oxides. The risk assessment indicated that for more than 50% of the samples the maximum daily intake exceeded the baseline value according to the EMA proposal. Nevertheless, the PA contents of *A. capillaris* samples from different sources varied greatly based on the HCA and PCA multivariate statistical method. Overall, the newly established UPLC-MS/MS method together with chemometrics could provide a new approach for the detection and risk assessment of toxic PAs in herbs, which would provide a meaningful reference for their rational utilization to ensure the public health safety.

## Figures and Tables

**Figure 1 molecules-24-01077-f001:**
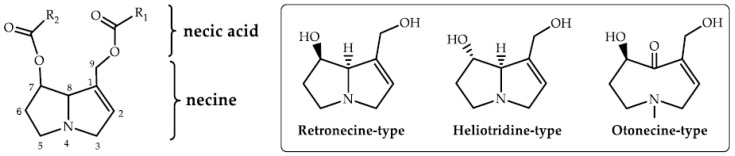
The chemical structures of toxic pyrrolizidine alkaloids. (**A**) is the basic skeleton and (**B**) is three main types of PAs.

**Figure 2 molecules-24-01077-f002:**
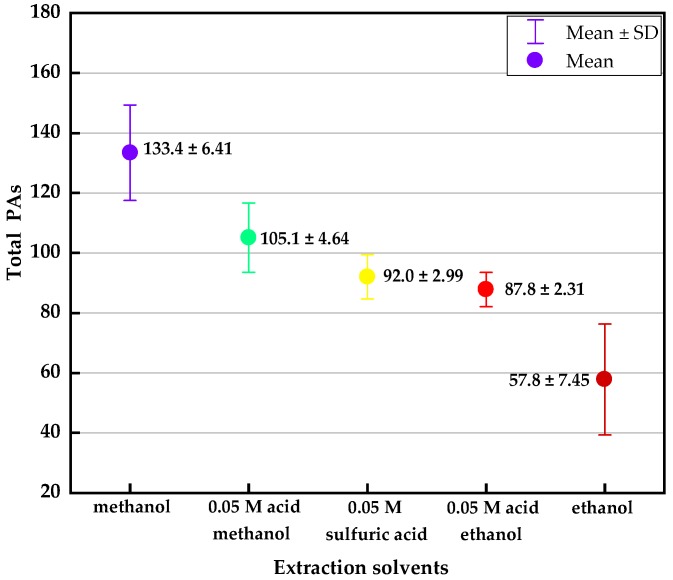
Effect of five different extraction solvents on the total PAs. Acid methanol: sulfuric acid in methanol, sulfuric acid: sulfuric acid water, acid ethanol: sulfuric acid in ethanol. The total PAs concentrations (mean ± SD, μg/kg) of different solvents are marked in the graph. SD: standard deviation, represents the deviation of three parallel samples with different solvents.

**Figure 3 molecules-24-01077-f003:**
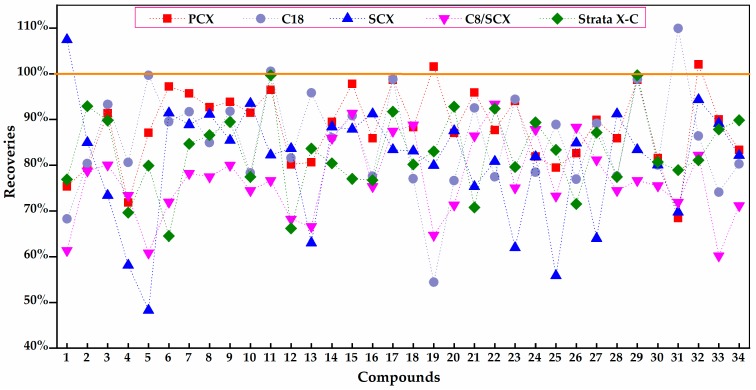
Effect of five different SPE cartridges on the recoveries of PAs. **1**: Retronecine, **2**: Echimidine, **3**: Echimidine N-oxide, **4**: Erucifoline, **5**: Erucifoline N-oxide, **6**: Europine, **7**: Europine N-oxide, **8**: Heliotrine, **9**: Heliotrine N-oxide, **10**: Intermedine, **11**: Intermedine N-oxide, **12**: Jacobine, **13**: Jacobine N-oxide, **14**: Lasiocarpine, **15**: Lasiocarpine N-oxide, **16**: Lycopsamine, **17**: Lycopsamine N-oxide, **18**: Monocrotaline, **19**: Monocrotaline N-oxide, **20**: Retrorsine, **21**: Retrorsine N-oxide, **22**: Senecionine, **23**: Senecionine N-oxide, **24**: Seneciphylline, **25**: Seneciphylline N-oxide, **26**: Senecivernine, **27**: Senecivernine N-oxide, **28**: Indicine, **29**: Indicine N-oxide, **30**: 7-Acetylintermedine, **31**: 7-Acetylintermedine N-oxide, **32**: Senkirkine, **33**: Trichodesmine, **34**: 7-Acetyllycopsamine.

**Figure 4 molecules-24-01077-f004:**
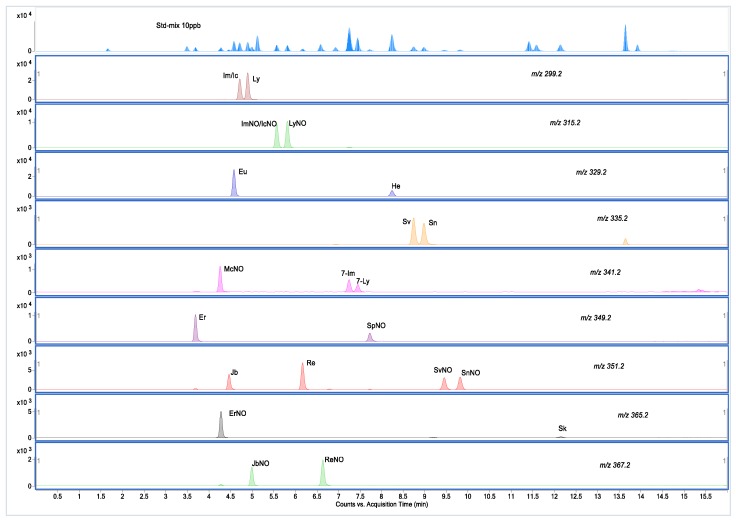
MRM chromatograms of nine sets of isomers with a mixed standard of PAs (10 μg/kg) by LC-MS/MS. For each pair the molecular mass and compound name abbreviations were shown.

**Figure 5 molecules-24-01077-f005:**
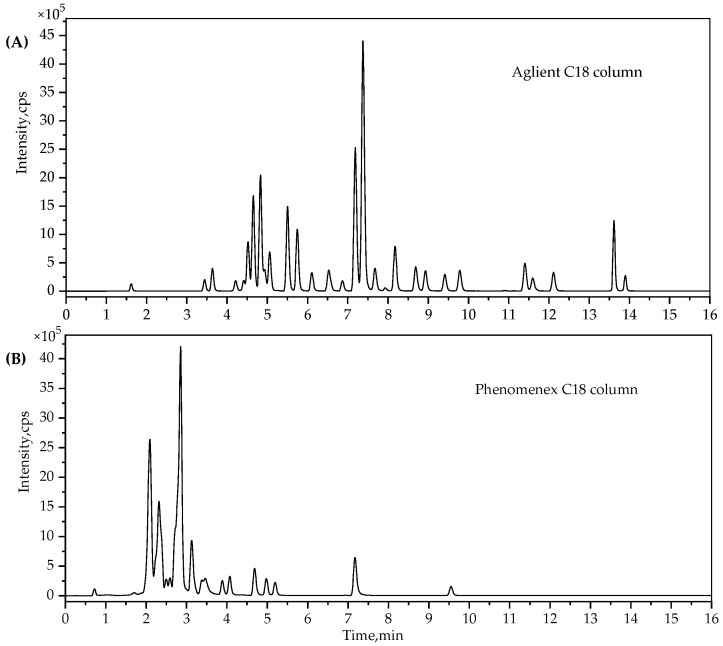
Total ion chromatogram (TIC) of two different kinds of columns. (**A**) is for the Agilent C_18_ column and (**B**) is for the Phenomenex C_18_ column, a mixed standard solution of 100 μg/kg PAs was used in this test.

**Figure 6 molecules-24-01077-f006:**
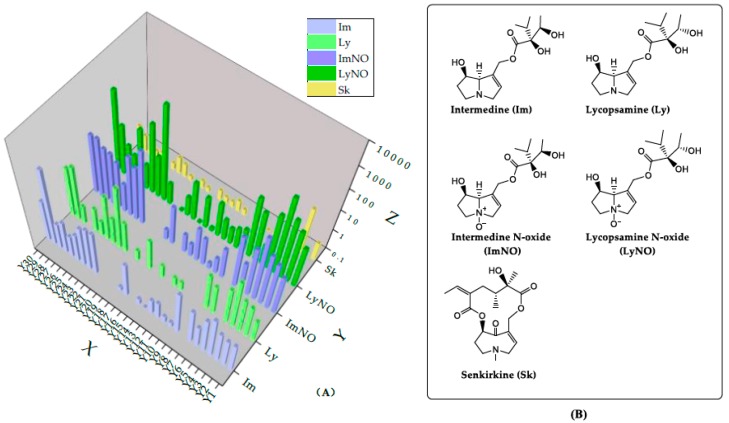
(**A**) The distribution of each major detected PA in *A. capillaris* samples. X: Batch number of samples, Y: detected major individual PA, Z: content of PAs in different samples. (**B**) the chemical structures of five detected PAs.

**Figure 7 molecules-24-01077-f007:**
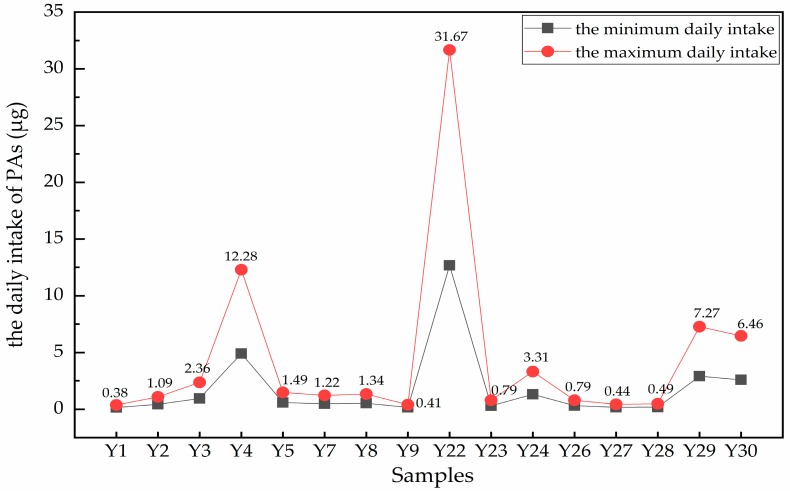
The maximum and the minimum daily intakes of *A. capillaris*. Only the 16 samples that displayed maximum daily intakes exceeding the baseline value (0.35 μg/day for a 50 kg person) are shown and their calculated values are marked on the chart.

**Figure 8 molecules-24-01077-f008:**
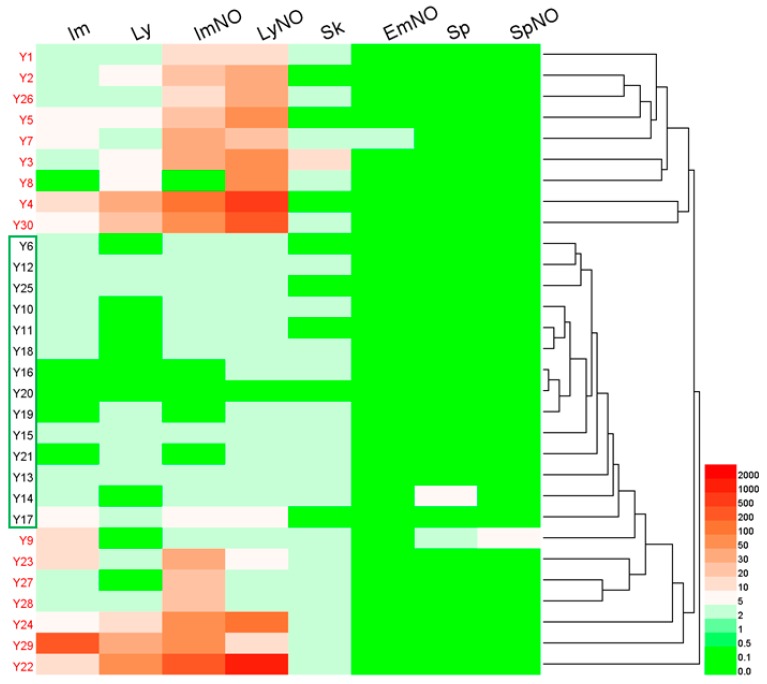
Hierarchical clustering heatmap of the detected PAs in *A. capillaris*. The total PAs concentration of the sample batches in red font exceeded the baseline, but the sample batches in black font and with green boxes did not.

**Figure 9 molecules-24-01077-f009:**
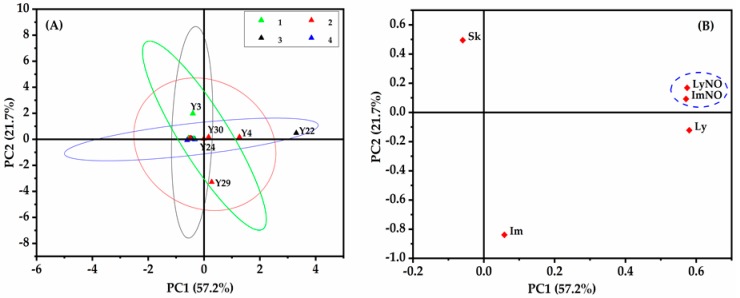
Principal component analysis of *A. capillaris* with the PAs daily intake above the baseline based on the contents of 5 main detected PAs analytes. (**A**) Score plots of PCA for 16 samples with different sources, where Group 1 are samples from North China, Group 2 from East China, Group 3 from the Northwest and Group 4 from Central-south and Southwest China; (**B**) loading plots of PCA for five main analytes to illuminate the relationship between PAs detected in *A. capillaris*.

**Table 1 molecules-24-01077-t001:** Limit of Detection (LOD), Limit of Quantification (LOQ), recoveries, Intra-day and Inter-day repeatability obtained by UPLC-MS/MS method.

No.	PAs Abbr.	LOD μg/kg	LOQ μg/kg	Recoveries (%) Mean ± SD, n = 3			Intra-Day RSD%, n = 6	Inter-Day RSD%, n = 6	Matrix Effects (%)
				1 μg/kg	10 μg/kg	100 μg/kg			
1	Ret	0.10	0.50	68.10 ± 0.35	75.38 ± 1.16	74.78 ± 2.12	4.54	3.32	47.22
2	Em	0.05	0.20	74.78 ± 0.18	79.41 ± 0.54	85.55 ± 1.16	0.94	5.04	96.91
3	EmNO	0.10	0.20	74.96 ± 0.21	91.38 ± 0.41	89.86 ± 0.78	1.51	7.64	96.48
4	Er	0.02	0.20	67.84 ± 1.55	71.82 ± 2.31	74.57 ± 1.02	0.59	4.37	90.16
5	ErNO	0.10	0.50	71.41 ± 1.24	87.15 ± 2.03	86.34 ± 1.43	1.78	3.27	102.36
6	Eu	0.01	0.10	73.33 ± 1.57	97.25 ± 1.14	91.47 ± 2.18	1.04	2.55	98.82
7	EuNO	0.02	0.10	82.25 ± 0.78	95.74 ± 2.24	97.40 ± 1.52	2.32	2.35	107.90
8	He	0.01	0.10	73.34 ± 0.35	92.67 ± 0.23	90.08 ± 0.80	0.65	2.81	103.98
9	HeNO	0.10	0.20	79.93 ± 0.46	93.88 ± 0.33	93.00 ± 2.25	1.86	3.76	95.06
10	Im	0.01	0.10	81.97 ± 1.34	91.45 ± 1.07	92.80 ± 1.11	0.79	4.72	87.77
11	ImNO	0.05	0.20	96.15 ± 1.47	96.50 ± 3.01	91.15 ± 2.38	6.88	4.04	114.20
12	Jb	0.10	0.50	77.63 ± 0.42	80.18 ± 1.12	78.82 ± 3.46	1.73	4.72	96.82
13	JbNO	0.10	0.50	71.64 ± 1.53	80.67 ± 2.45	81.99 ± 3.12	3.99	3.71	88.31
14	Lc	0.01	0.05	71.75 ± 0.57	89.46 ± 0.53	84.90 ± 5.12	1.89	4.48	100.80
15	LcNO	0.05	0.20	73.70 ± 0.95	97.82 ± 2.58	91.51 ± 0.31	2.43	5.05	97.42
16	Ly	0.01	0.10	76.47 ± 0.07	85.91 ± 1.65	89.52 ± 1.26	3.32	4.45	94.85
17	LyNO	0.05	0.2	101.58 ± 1.43	98.71 ± 1.53	93.08 ± 2.03	2.81	3.23	93.07
18	Mc	0.02	0.10	74.01 ± 1.14	88.38 ± 2.14	89.62 ± 1.68	0.38	4.35	91.92
19	McNO	0.20	0.50	84.62 ± 0.50	101.59 ± 1.09	92.46 ± 1.34	2.30	2.86	100.10
20	Re	0.05	0.50	78.97 ± 0.78	87.07 ± 0.55	88.80 ± 4.61	2.11	5.09	92.57
21	ReNO	0.10	0.50	74.40 ± 0.62	95.93 ± 2.32	86.89 ± 6.07	2.42	6.87	91.56
22	Sn	0.10	0.50	71.52 ± 0.85	87.74 ± 0.76	84.53 ± 7.43	1.24	1.10	106.26
23	SnNO	0.10	0.50	73.22 ± 0.98	94.08 ± 1.04	89.74 ± 0.87	0.47	3.58	108.66
24	Sp	0.10	0.50	69.65 ± 1.34	81.92 ± 4.34	80.32 ± 2.24	1.36	3.51	105.05
25	SpNO	0.20	0.50	73.51 ± 1.42	79.47 ± 1.21	80.19 ± 1.05	2.44	4.20	106.86
26	Sv	0.05	0.20	83.60 ± 0.25	82.63 ± 1.11	82.90 ± 1.14	0.75	2.52	103.74
27	SvNO	0.20	0.50	78.02 ± 0.34	89.94 ± 2.53	90.67 ± 5.21	4.29	3.27	112.64
28	Ic	0.01	0.10	81.97 ± 1.34	91.45 ± 1.07	92.80 ± 1.11	0.79	4.72	87.77
29	IcNO	0.05	0.20	96.15 ± 1.47	96.50 ± 3.01	91.15 ± 2.38	6.88	4.04	114.20
30	7-Im	0.01	0.10	70.00 ± 1.52	81.60 ± 1.26	82.67 ± 1.26	1.17	0.14	110.29
31	7-ImNO	0.05	0.20	70.45 ± 1.29	71.42 ± 0.94	71.52 ± 0.75	4.42	4.50	93.34
32	Sk	0.02	0.10	102.96 ± 3.07	100.08 ± 2.35	90.05 ± 1.56	1.19	1.43	97.50
33	Td	0.05	0.20	70.62 ± 0.24	90.04 ± 0.04	84.26 ± 0.17	3.33	3.95	101.22
34	7-Ly	0.01	0.10	70.65 ± 1.53	83.40 ± 1.48	80.49 ± 0.04	0.45	1.04	95.62

**Table 2 molecules-24-01077-t002:** The frequency and content range of individual PAs in *A. capillaris* samples.

PAs	Im	Ly	ImNO	LyNO	Sk	EmNO	Sp	SpNO
Positive Samples	25	22	25	29	22	1	2	1
Range(μg/kg)	0.11-383.28	0.29-92.05	0.20-255.46	0.21-1750.99	0.10-15.81	3.25	1.35-5.14	5.57

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
