# Peer review of "Simultaneous Determination and Risk Assessment of Pyrrolizidine Alkaloids in Artemisia capillaris Thunb. by UPLC-MS/MS Together with Chemometrics"

_molecules, 2019, doi:10.3390/molecules24061077_

Round 1
Reviewer 1 Report
02- March -2019
Journal: Molecules
Manuscript ID: molecules-454519
Title: Simultaneous Determination and Risk Assessment of Pyrrolizidine Alkaloids in Artemisia capillaris Thunb. By UPLC-MS/MS Together with Chemometrics
Authors: Li-Hua Chen , Jun-Chi Wang , Qi-Lei Guo , Yue Qiao , Hui-Juan Wang , Yong-Hong Liao ,Di-An Sun and Jian-Yong Si
Dear Editor:
The authors have investigated the Pyrrolizidine Alkaloids in Artemisia capillaris. This is a good comprehensive manuscript using the chemometrics technological tool. The explanation of this study design is reasonable and thus, in my opinion the manuscript is adequate for publication after some major changes as indicated below.
Comments to Authors
Abstract:
1- The authors should add high quality graphical abstract in accordance to the journal instructions.
2- Using chemometrics together with UPLC-MS/MS is a good and new idea
3- “and the further risk evaluation would provide a guiding basis for its safe medication.” What do you mean?
Keywords:
The authors could minimize the number of keywords
Introduction:
1- Pyrrolizidine alkaloids (PAs), add the main skeleton
2- “PAs are mainly distributed in these three families: Asteraceae, Boraginaceae and Fabaceae” please give examples
3- The authors shouldn`t repeat (and) as in lines: 38, 46, 47 and 63.
4- The authors would clarify this sentence: (Dutch recommended a tolerable daily intake for PAs 50 of 1 μg/kg b.w. and 0.1 μg/kg b.w. per day, respectively) in line 51.
5- Please explain COT, BfR studies and SPE cartridges
6- “To date, high levels of PAs have been reported in many herbal drugs”, please give examples and explain in more details the acute toxicity?
7- In line 73: It is not good to start a sentence with (and).
8- The authors should mention the other sources of PAS other than plants
9- “It is expected that the developed and validated UPLC-MS/MS approach and optimized pretreatment method could be used for the determination of A. capillaris containing PAs, and it would provide safety evidence for its rational drug utilization based on the risk assessment and statistical chemometrics analysis” this statement is not clear enough
10- Introduction should be rewritten in a proper way, it is very confusing
11- The idea of the paper is good but the authors should explain the aim in more details
Results and Discussion:
1- The authors would clarify this part (Pretreatment Method Development) in a brief way.
2- The authors would check the sentences without references.
3- Please explain figure 1 in more details
4- figure 2: will be more informative if the authors discuss the result.
5- From line 180-183: the authors would explain the main indication of the number range of LOD and LOQ between (0.01~0.2 μg/kg and 0.1~0.5 μg/kg for 34 Pas)
6- Taken together, the author didn`t discuss the results clearly.
Material and Methods:
1- Material and Methods seem to be suitable for the objectives of this study.
2- “A. capillaris samples (n=30) deriving from 21 different provinces were purchased from several medicine markets and pharmacies.” which part of the plant?. The authors would get samples from local farms
3- The authors didn`t add figure to show the chemical structures .
4- If there are supplementary table such as (Table S1,S3, Table S4) in line 361,369 the authors would add these material.
5- The authors would identify the (Agilent Jet Stream technology).
6- Please add references to the material and methods section
7- The authors could benefit from the following reference in the material and method section:
Farag, M., Ali, S., Hodaya, R., El-Seedi, H., Sultani, H., Laub, A., Eissa, T., Abou-Zaid, F. and Wessjohann, L., 2017. Phytochemical profiles and antimicrobial activities of Allium cepa red cv. and A. sativum subjected to different drying methods: a comparative MS-based metabolomics. Molecules, 22(5), p.761.
Author Response
Response for molecules-454519
Manuscript ID: molecules-454519
Title: Simultaneous Determination and Risk Assessment of Pyrrolizidine Alkaloids in Artemisia capillaris Thunb. by UPLC-MS/MS Together with Chemometrics
Authors: Li-hua Chen, Jun-chi Wang, Qi-Lei Guo, Yue Qiao, Hui-Juan Wang, Yong-Hong Liao, Di-An Sun, Jian-yong Si *
Dear Editors and Reviewers,
Thank you very much for giving us the opportunity to revise our manuscript. we would like to express our great appreciations for your warm work and the reviewers’ constructive suggestions with regard to our manuscript.
We have studied the comments carefully and tried our best to improve the manuscript according to the reviewers’ suggestions. Any revisions were clearly highlighted using the "Track Changes" function. All the lines and pages indicated above are in the revised manuscript. The responses to the comments and the main corrections in the paper are listed below point by point.
I hope these corrections could meet with approvals and this manuscript could be published in Molecules.
Please feel free to contact us if you have any questions.
Best wishes!
Sincerely yours,
Jian-Yong Si
Email: jysi@implad.ac.cn
Institute of Medicinal Plant Development, Chinese Academy of Medical Sciences & Peking Union Medical College, Beijing, China
Response to Reviewer 1 Comments
Abstract
1: The authors should add high quality graphical abstract in accordance to the journal instructions.
Response 1: “images displayed online will be up to 11 by 9 cm on screen” according to the requirements of Molecules, and it was compressed in the PDF file you reviewed, so the picture was obscure. For this reason, I uploaded a single “graphical abstract” tif file with 300 dpi, but it was still compressed in the PDF file, so I re-uploaded a graphic abstract with high quality PDF format (300 dpi). I hope you can see the newly uploaded graphical abstract. Thanks for your recommendation.
2: Using chemometrics together with UPLC-MS/MS is a good and new idea.
Response 2: Thank you for your recognition of our idea and for taking your valuable time to read this manuscript.
3: “and the further risk evaluation would provide a guiding basis for its safe medication.” What do you mean?
Response 3: We have made a preliminary risk assessment based on the European Medicines Agency guideline concerning the daily intake of PAs, and it is expected to provide safety evidence for the rational use of Artemisia capillaris Thunb. I am sorry for not making this statement clearly in the abstract. I have rewritten the sentence – “The method was successfully applied to the detection and risk evaluation of PAs-containing A. capillaris for the first time. This study will provide a meaningful reference for the rational and safe use of A. capillaris”. Thank you for your kind comments and the information you provided.
Keywords: The authors could minimize the number of keywords.
Response 4: Thank you for your valuable advice. The keywords " hierarchical clustering analysis " and " principal component analysis" which belong to the chemometrics methods have been deleted.
Introduction
1: Pyrrolizidine alkaloids (PAs), add the main skeleton
Response 5: Thank you for your careful and precise work. I have added the main skeleton of PAs in page 2, line 104-106. Furthermore, the chemical structures of three major types of PAs were added.
2: “PAs are mainly distributed in these three families: Asteraceae, Boraginaceae and Fabaceae” please give examples
Response 6: PAs are found predominantly in some tribes or genera of the families Asteraceae, Boraginaceae and Fabaceae within traditional Chinese medicines. “and of the Asteraceae, the majority exist in the tribes Senecioneae, Eupatorieae and Ageratum” (page 2, line 109 ), such as Senecio Scandens, Eupatorium cannabinum and Ageratum conyzoides. Besides, the other PAs-containing plants are the genus Heliotropium (e.g. Heliotropium indicum), Cynoglossum (e.g. Cynoglossum officinale) and Lithospermum (e.g. Lithospermum erythrorizon) of the family Boraginaceae and the genus Crotalaria (e.g. crotalaria sessiiflora) of Fabaceae family. PAs isolated from three families mentioned in the manuscript are summarized in several review articles and books, such as “Roeder, E. Medicinal plants in China containing pyrrolizidine alkaloids. Pharmazie 2000, 55, 711-726” and “Tamariz, J.; Burgueño-Tapia, E.; Vázquez, M. A.; et al. Chapter One-Pyrrolizidine Alkaloids. In The Alkaloids: Chemistry and Biology, Academic Press: Pittsburgh, USA, 2018, Vol. 80, pp 1-314”.
According to your comment, I have added three notable used PAs-containing plants of the Asteraceae family. Thank you very much.
3: The authors shouldn`t repeat (and) as in lines: 38, 46, 47 and 63.
Response 7: Thank you for your careful reading of our manuscript, I have revised these sentences according to your recommendation. They were now in lines: 36, 109, 131.
4: The authors would clarify this sentence: (Dutch recommended a tolerable daily intake for PAs 50 of 1 μg/kg b.w. and 0.1 μg/kg b.w. per day, respectively) in line 51.
Response 8: I am sorry for our fault statement of this sentence. “The Australia and the Dutch recommended a tolerable daily intake for PAs of 1 μg/kg b.w. and 0.1 μg/kg b.w. per day, respectively.” This sentence is intended to express that the recommended daily intake of PAs in Australia is 1 μg/kg body weight and the Netherlands is 0.1 μg/kg body weight. According to your comments, I have rephrased this sentence in page 2, line 142-143.
5: Please explain COT, BfR studies and SPE cartridges
Response 9: “COT” is the abbreviation of “The UK Committee on Toxicity of Chemicals in Food, Consumer Products and the Environment” (page , line ), “BfR” is the abbreviation of “The German Federal Institute for Risk Assessment” (page 2, line 144) and SPE cartridges refers to solid phase extraction columns (page 2, line 146). I used abbreviations to avoid redundant words and listed their full names when first written. Thank you for your reminding.
6: “To date, high levels of PAs have been reported in many herbal drugs”, please give examples and explain in more details the acute toxicity?
Response 10: (1) Bodi D et al. revealed a maximum PAs level of 3099 μg/kg in anise (reference 18: Bodi, D.; Ronczka, S.; Gottschalk, C.; et al. Determination of pyrrolizidine alkaloids in tea, herbal drugs and honey. Food Addit. Contam. 2014, 31, 1886-1895). Avula B et al. found that the total contents of PAs were up to 1991 μg/g in Senecio riddellii and were 3240 μg/kg in Eupatorium cannabinum (reference 19: Avula, B.; Sagi, S.; Wang, Y.-H.; et al. Characterization and screening of pyrrolizidine alkaloids and N-oxides from botanicals and dietary supplements using UHPLC-high resolution mass spectrometry. Food Chem. 2015, 178, 136-148). The average total PAs concentration of Emilia sonchifolia dry herb were reported from a low of 33.3 μg/g to a high of 93.9 μg/g (reference 20: Hsieh, C.-H.; Chen, H.-W.; Lee, C.-C.; et al. Hepatotoxic pyrrolizidine alkaloids in Emilia sonchifolia from Taiwan. J. Food Compos. Anal. 2015, 42, 1-7). I have cited references at the end of this sentence (page 2, line 112).
(2) In acute animal toxicological experiments, the LD50s of pyrrolizidine alkaloids in rats varies from 34 to 300 mg/kg, with some approaching 1,000 mg/kg. Hepatic necrosis is most predominant symptom in acute poisoning and severe liver damage is the cause of death. Animals died successively within about 7 days after exposure to large amounts of PAs. In human acute poisoning, Hepatic veno-occlusive disease, characterized by abdominal pain and distension due to ascites, is associated with human consumption of PAs-contaminated foods or herbs.
7: In line 73: It is not good to start a sentence with (and) .
Response 11: Thank you for your kind advice. I have rewritten this sentence.
8: The authors should mention the other sources of PAS other than plants
Response 12: In addition to plants, it has been reported that honey and pollen are also important sources of PAs. However, pollen is a part of plant and the presence of PAs in honey has also been shown to be associated with floral sources of PAs-containing plants (reference: Beales K A , Betteridge K , Colegate S M , et al. Solid-Phase Extraction and LC−MS Analysis of Pyrrolizidine Alkaloids in Honeys[J]. Journal of Agricultural and Food Chemistry, 2004, 52(21):6664-6672).
9: “It is expected that the developed and validated UPLC-MS/MS approach and optimized pretreatment method could be used for the determination of A. capillaris containing PAs, and it would provide safety evidence for its rational drug utilization based on the risk assessment and statistical chemometrics analysis” this statement is not clear enough
Response 13: I am sorry for not making this statement clear enough. I have rewritten this sentence (page 3, line 236-239) according to your comment. Thank you for pointing out the deficiencies in the statement.
10: Introduction should be rewritten in a proper way, it is very confusing
Response 14: I am deeply sorry for not giving a clear background and expressing my purpose clearly. I have rewritten the introduction carefully according to your suggestion. In order to guarantee logical coherence and topic highlight, the order of some paragraphs were changed, some sentences were rivised. I did not list the changes here but marked them in red in red in the word version of revised manuscript. Thank you for your instructive advice.
11: The idea of the paper is good but the authors should explain the aim in more details
Response 15: The aim of this study was to establish a sensitive and efficient UPLC-MS/MS and chemometrics method for the simultaneous determination and risk assessment of PAs in A. capillaris, which will provide important information to the public for the rational and safe use of A. capillaris.
Results and Discussion:
1: The authors would clarify this part (Pretreatment Method Development) in a brief way.
Response 16: I have re-discussed this part (Pretreatment Method Development) in a brief and clear way. Thank you for your comment.
2: The authors would check the sentences without references.
Response 17: I have checked the sentences and have deleted some references according to your suggestion.
3: Please explain figure 1 in more details
Response 18: Thanks for your kind comment. I have rewritten the Section 2.1.1. All the changes were marked red.
4: figure 2: will be more informative if the authors discuss the result.
Response 19: According to your kind suggestion, I have rewritten the Section 2.1.2. All the changes were marked red. Thank you very much.
5: From line 180-183: the authors would explain the main indication of the number range of LOD and LOQ between (0.01~0.2 μg/kg and 0.1~0.5 μg/kg for 34 Pas)
Response 20: I am sorry for not explaining the range of LOD (limits of detection) and LOQ (limits of quantification). 0.01~0.2 μg/kg means “the LOD of 34 PAs are ranging from 0.01 to 0.2 μg/kg” . 0.1~0.5 μg/kg means “the LOQ of 34 PAs are ranging from 0. 1 to 0.5 μg/kg”. The specific LOD and LOQ of each PA are shown in table 2. I have added the full names of LOD and LOQ (marked in red). Thank you for your suggestion.
6:Taken together, the author didn`t discuss the results clearly.
Response 21: Thank you for pointing out the issues in the Section results and discussion, I have had a serious discussion about the results. Many of the additions are highlighted in red.
Material and Methods:
1: Material and Methods seem to be suitable for the objectives of this study.
Response 22: Thank you for your comment and for your careful reading of our manuscript
2: “A. capillaris samples (n=30) deriving from 21 different provinces were purchased from several medicine markets and pharmacies.” which part of the plant?. The authors would get samples from local farms
Response 23: (1) the dry aerial part were selected of A. capillaris for experiment.
(2) Because this herb is widely sold in pharmacies and medicine markets, it is also used directly as medicine by people, so we bought different batches of A. capillaris from some large pharmacies such as Tongrentang and herbal medicine markets in Bozhou city. All of them have been identified by professionals who specialize in plant resources. Thank you for your kind suggestion.
3: The authors didn`t add figure to show the chemical structures.
Response 24: The chemical structures of 34 PAs were shown in Figure S2 of “Supplementary Files”, because there are many structures, which would take up a large part of the article. That would be great if you could see the supplementary materials in “Supplementary Files”.
4: If there are supplementary table such as (Table S1,S3, Table S4) in line 361,369 the authors would add these material.
Response 25: I am very sorry to hear that you did not see the supplementary tables. I have re-uploaded “Supplementary Files” and hope they do not affect your reading.
5: The authors would identify the (Agilent Jet Stream technology).
Response 26: Yes. Compared to conventional electrospray ionization (ESI), the Agilent Jet Stream technology enables sensitivity improvements of about 10-fold in negative ion mode and between 4-to-10-fold in positive ion mode. More details are available on https://www.agilent.com/cs/library/technicaloverviews/public/5990-3494en_lo%20CMS.pdf I have added this article in the references. Thank you for your reminder.
6: Please add references to the material and methods section
Response 27: I have added two references in the material and methods section according to your suggestion.
7: The authors could benefit from the following reference in the material and method section:
Farag, M., Ali, S., Hodaya, R., El-Seedi, H., Sultani, H., Laub, A., Eissa, T., Abou-Zaid, F. and Wessjohann, L., 2017. Phytochemical profiles and antimicrobial activities of Allium cepa red cv. and A. sativum subjected to different drying methods: a comparative MS-based metabolomics. Molecules, 22(5), p.761.
Response 28: I have added the above reference in the material and methods section. Thank you for your recommendation.
Once again, thank you very much for your comments and suggestions.
Other corrections
1. Graph modification
Figure 2: To explain the extraction solvents more accurate, the expression “95% CI” was replaced with “SD”. Each test with different extraction solvents was repeated three times in parallel, SD is standard deviation.
Figure 3: For the sake of labeling, the PAs abbreviations of the X-axis were replaced with numbers, they were all explained under the figure.
Figure 6: To explain the work more clearly, we added the chemical structures of 5 detected PAs in most A. capillaris.
Table 1: In order to guarantee accuracy of expression, the word “PAs” (in the second column, the first row) was changed to “PAs Abbr.”.
Table S3: In order to keep the article concise, Table 3 in the previous manuscript was put in Table S3 of the supplementary material which are available online.
2.Words modification
To improve the manuscript, we have made some corrections on some words in the manuscript. These changes were marked in red of revised manuscript.
3. Sequence adjustment of paragraphs and graph
(1) In order to guarantee sequential coherence, the order of second (In order to summarize……) and third paragraph (The concentration of……) of Page 8 were changed.
(2) To make the structure of the manuscript more coherent, The placement location of several graphs have been adjusted.

Reviewer 2 Report
Dear Authors,
having read your manuscript I have some doubts about the shape of your manuscript.
It is interesting, that you managed to identify a wide spectrum of PAs in the extracts of A. capillaris.
In my opinion this manuscript should be developed in a way to show a very close relation to other studies.
PAs are already widely studied In your manuscript you present your results, but do not discuss with the results of other researchers - which columns did other authors use, how many minutes their run lasted, which components they did identify, what was their method of extraction.
Like this, this manuscript presents some novelty, but as it ontains a lot of data on the optimisation of methodology and analytics of a group of component, which are already widely studied, the results need to be further discussed.
Other remarks:
- Abstract: 'remains a mystery' is not an bjectiv neutral expression - please, rewrite the sentence
-line 40: are there any non-toxic PA's? please, comment on that
- I would recommend to move the introdution to the toxicity of PAs from the line 44 to the line 40 to introduce the readers into the subject better
-lines 60 and 64: change 'was' to 'were'
- rewrite line 73. do not tart with 'and'
-lines 83 and on - these are data suitable for the discussion - here describe, hat was the aim of the study
- in the introduction mention the hypotheses which state, that the presence of PAs in the plant samples is related to the coexisting with a plant funghi or bateria
-line 113 - how do your result correspond to the results of other researchers?
- figure 2 - the used abbreviations should be explained under the figure
- show the chromatograms obtained by the aplication of different columns in the manusript body
- prepare a table with detailed MS and MS/MS data - the theoretical masses, measured masses,
DBE values, error of measurement, chemical formula
- how did the authors select the tools for sample preconcentrtion - why did they select given adsorbents?
- prepare a table showing quantitative differences in the concentrations of all alkaloids depending on the extractant used.
Author Response
Response for molecules-454519
Manuscript ID: molecules-454519
Title: Simultaneous Determination and Risk Assessment of Pyrrolizidine Alkaloids in Artemisia capillaris Thunb. by UPLC-MS/MS Together with Chemometrics
Authors: Li-hua Chen, Jun-chi Wang, Qi-Lei Guo, Yue Qiao, Hui-Juan Wang, Yong-Hong Liao, Di-An Sun, Jian-yong Si *
Dear Editors and Reviewers,
Thank you very much for giving us the opportunity to revise our manuscript. we would like to express our great appreciations for your warm work and the reviewers’ constructive suggestions with regard to our manuscript.
We have studied the comments carefully and tried our best to improve the manuscript according to the reviewers’ suggestions. Any revisions were clearly highlighted using the "Track Changes" function. All the lines and pages indicated above are in the revised manuscript. The responses to the comments and the main corrections in the paper are listed below point by point.
I hope these corrections could meet with approvals and this manuscript could be published in Molecules.
Please feel free to contact us if you have any questions.
Best wishes!
Sincerely yours,
Jian-Yong Si
Email: jysi@implad.ac.cn
Institute of Medicinal Plant Development, Chinese Academy of Medical Sciences & Peking Union Medical College, Beijing, China
Response to Reviewer 2 Comments
Point 1: It is interesting, that you managed to identify a wide spectrum of PAs in the extracts of A. capillaris. In my opinion this manuscript should be developed in a way to show a very close relation to other studies.
Response 1: Thank you for your instructive suggestion. This study combined UPLC-MS/MS with chemometrics method for the determination and risk assessment of PAs in A. capillaris, it is expected this method could be extended to the detection and analysis of other PAs- containing herbs.
Point 2: PAs are already widely studied in your manuscript you present your results, but do not discuss with the results of other researchers - which columns did other authors use, how many minutes their run lasted, which components they did identify, what was their method of extraction.
Response 2: Thank you very much for point out the results and discussion issues. According to your suggestion, to explain the results in a brief way, I have deleted these contents - “Bharathi Avula et al ……”, “According to the BfR method protocol……”, “Gradient elution conditions……” and so on.
Point 3: Like this, this manuscript presents some novelty, but as it contains a lot of data on the optimisation of methodology and analytics of a group of component, which are already widely studied, the results need to be further discussed.
Response 3: According to your suggestion, I have further discussed the results regarding the optimisation of methodology and analytics of PAs in Section 2.1 and Section 2.2. All the further discussion were marked red. Thank you very much.
Point 4: - Abstract: 'remains a mystery' is not an objective neutral expression - please, rewrite the sentence
Response 4: Thank you for your valuable advice. I have rewritten the sentence according to your suggestion.
Point 5: -line 40: are there any non-toxic PA's? please, comment on that
Response 5: Yes. PAs are not all toxic. In general, saturated PAs is non-toxic, that is, there is no double bond between the position C1 and C2, such as platyphylline (Molecular Formula: C18H27NO5 ) isolated from Senecio oryzetorum (reference: Ruan J; Liao C; Ye Y; et al. Lack of Metabolic Activation and Predominant Formation of an Excreted Metabolite of Nontoxic Platynecine-Type Pyrrolizidine Alkaloids[J]. Chemical Research in Toxicology, 2014, 27(1):7-16).
Point 6: - I would recommend to move the introduction to the toxicity of PAs from the line 44 to the line 40 to introduce the readers into the subject better
Response 6: Thank you for your instructive recommendation. To make subject highlight and logical coherence, I have rewritten the introduction and first introduced the toxicity of PAs. All the changes were marked in red.
Point 7: -lines 60 and 64: change 'was' to 'were'
Response 7: I am sorry for these verb mistakes. I have revised these sentences. Furthermore, I have had the manuscript checked by several native English speaking colleagues. Thank you for your careful reading.
Point 8: - rewrite line 73. do not start with 'and'
Response 8: I have rewritten this sentence according to your advice. Thank you very much.
Point 9: -lines 83 and on - these are data suitable for the discussion - here describe, what was the aim of the study
Response 9: Thanks for your kind recommendation. I have rewritten these sentences in page 3, line 228-230. The aim of this study was to establish a sensitive and efficient UPLC-MS/MS and chemometrics method for the simultaneous determination and risk assessment of PAs in A. capillaris, which will provide important information to the public for the rational and safe use of A. capillaris.
Point 10: - in the introduction mention the hypotheses which state, that the presence of PAs in the plant samples is related to the coexisting with a plant funghi or bateria
Response 10: PAs are secondary metabolites of plants, however, whether the presence of PAs in the plant samples is related to the coexisting with a plant funghi or bateria needs to be further studied.
Point 11: -line 113 - how do your result correspond to the results of other researchers?
Response 11: The extraction condition is sonication with methanol in the reported study (reference: Avula, B.; Sagi, S.; Wang, Y.-H.; et al. Characterization and screening of pyrrolizidine alkaloids and N-oxides from botanicals and dietary supplements using UHPLC-high resolution mass spectrometry. Food Chem. 2015, 178, 136-148).
Point 12: - figure 2 - the used abbreviations should be explained under the figure
Response 12: Thank you for your careful work. I am sorry for not explaining the used abbreviations. I have added the full name of each PA under the figure. Besides, in order to To make it easier to label, the PAs abbreviations of the X-axis were replaced with numbers.
Point 13: - show the chromatograms obtained by the application of different columns in the manuscript body
Response 13: I have added the chromatograms obtained by the application of different columns in Figure 5 according to your suggestion.
Point 14: - prepare a table with detailed MS and MS/MS data - the theoretical masses, measured masses, DBE values, error of measurement, chemical formula
Response 15: I am sorry to hear that you did not see the table. Given that it would take up a large part of the article, I have uploaded detailed MS and MS/MS data (including measured masses, retention times, product ions, fragmentor and collision energy) of 34 PAs in Table S2 of “Supplementary Files”. Due to the low resolution of the instrument, the theoretical masses were basically consistent with the measured masses, so we did not add the theoretical masses, measured masses, DBE values and error of measurement. I have re-upload these supplementary materials, that would be great if you could see them. The chemical formula of each PA was shown in Figure S2.
Point 15: - how did the authors select the tools for sample preconcentration - why did they select given adsorbents?
Response 15: We considered two factors: the total extraction concentration and recoveries of PAs in A. capillaris. Finally, we selected the given adsorbents - Cleanert PCX SPE, a mixed-mode strong cation exchange sorbent. Because it can provides dual retention modes of reversed-phase and cation-exchange. It has high surface area and a wide usable pH range of 0-14.
Point 16: - prepare a table showing quantitative differences in the concentrations of all alkaloids depending on the extractant used.
Response 16: I have added the table (given in Table S1-2 of “Supplementary Files” ) showing the quantitative differences in alkaloid concentrations of all the different extractants used according to your suggestion. Furthermore, I have remarked the average total concentrations of PAs in Figure 2.Thank you for your instructive suggestion.
Once again, thank you very much for your comments and suggestions.
Other corrections
1. Graph modification
Figure 2: To explain the extraction solvents more accurate, the expression “95% CI” was replaced with “SD”. Each test with different extraction solvents was repeated three times in parallel, SD is standard deviation.
Figure 3: For the sake of labeling, the PAs abbreviations of the X-axis were replaced with numbers, they were all explained under the figure.
Figure 6: To explain the work more clearly, we added the chemical structures of 5 detected PAs in most A. capillaris.
Table 1: In order to guarantee accuracy of expression, the word “PAs” (in the second column, the first row) was changed to “PAs Abbr.”.
Table S3: In order to keep the article concise, Table 3 in the previous manuscript was put in Table S3 of the supplementary material which are available online.
2.Words modification
To improve the manuscript, we have made some corrections on some words in the manuscript. These changes were marked in red of revised manuscript.
3. Sequence adjustment of paragraphs and graph
(1) In order to guarantee sequential coherence, the order of second (In order to summarize……) and third paragraph (The concentration of……) of Page 8 were changed.
(2) To make the structure of the manuscript more coherent, The placement location of several graphs have been adjusted.

Round 2
Reviewer 1 Report
I recommend the paper for publication. The authors have considered carefully all comments.
Reviewer 2 Report
Dear Authors
Thank you very much indeed for all extensive corrections which you introduced to the body of your manuscript